# The Identification Method of the Winding Vibration Faults of Dry-Type Transformers

**Shulian Liu** [1,*]**, Ling Zhang** [1]**, Likang Yang** [1]**, Cunkai Gu** [1] **and Zaihua Wang** [2]

1   Department of Mechanical and Energy Engineering, Zhejiang University of Science and Technology, Hangzhou 310023, China
2   Research Institute of Electric Power Academy of State Grid Zhejiang Electric Power Company, Hangzhou 310014, China
*   Correspondence: 104041@zust.edu.cn

**Abstract:** To identify the four typical faults of dry-type transformer winding insulations, looseness, deformation and eccentricity, this study establishes the electric magnetic force multi-physical field simulation model of a dry-type transformer winding under the four typical faults with COMSOL software, based on the vibration mechanism of an SCB10-1000/10 dry-type transformer. Through the multi-physical field coupling calculation, the comparative relationship between the vibration acceleration of the winding under the four kinds of faults and the normal working state is obtained. The results show that the amplitude growth rate of the fundamental frequency or harmonic frequency of the acceleration signal under four kinds of faults is different from that under normal conditions. Therefore, the threshold value of the fundamental frequency or harmonic increment of the acceleration signal is introduced to describe the growth rate of the acceleration signal relative to normal conditions. Finally, four typical faults are identified with different threshold ranges of acceleration increment under faults, laying a foundation for the fault diagnosis of transformer winding vibrations.

**Keywords:** transformer winding; multiple physical fields; mechanical failure; vibration

## 1. Introduction

The transformer is a critical piece of equipment in the power grid, and its security and reliability are central to the normal operation of the whole power grid. Due to their small size, efficiency, quietness, low maintenance requirements, and other advantages, dry-type transformers are often used in situations with high-performance requirements. These might include fire or explosion protection situations, as well as some power supply and distribution locations [1]. The high failure rate of transformers affects the safe and stable operation of the power grid, and the mechanical failure of transformers is often caused by potential problems not being proactively addressed. An important cause of transformer failure [2–6] is the vibration of the windings and iron cores. The transformer is mainly composed of an iron core and a winding. Once the iron core or winding is damaged, the use of the transformer would be seriously affected. The quality of the winding has always been one of the important standards for the efficient operation of the transformer [7]. According to statistics, transformer faults caused by iron cores and windings rank third among all faults, which is also the main source of transformer faults [8–11]. The causes of winding faults are complex. The winding element is critical to transformer design because the number of turns in the primary and secondary windings determines the transformer's voltage ratio. If a transformer's winding mechanism fails, the results include overload, ageing, surges, and deterioration over time, which could lead to problems with the mechanism, including loosening, insulation and distortion. Of these, surges caused by lightning and switching could be transferred from one voltage level to another via transformer coupling. In turn, however, this could lead to failures caused by dielectric stress. An overload of short-circuit currents to the transformer could also cause it to malfunction, and it is, therefore, important

to review the inrush current forces and short circuit forces [12–14]. In addition, the analysis of the vibration levels of the transformer windings is critical to diagnosing the faults within a transformer. Wu et al. used the ANSYS Workbench software to simulate the short-circuit and DC magnetic bias conditions on the secondary side of the transformer and extract the vibration characteristics of transformers under different conditions, providing a basis for transformer fault diagnosis [15]. Wu et al. created a diagnostic method for transformer winding based on three parameters: winding resistance, leakage induction, and capacity. The online identification of winding parameters was achieved with a recursive least squares algorithm, and the classification of the winding deformation and the diagnosis of the deformation degree was achieved by using the online identified winding parameters, and the results showed that the method had a high overall accuracy [16]. Zhang et al. used MAXWELL to carry out an electromagnetic simulation analysis on a transformer winding and obtained the electromagnetic density cloud diagram of the winding iron core under the primary rated voltage, induced voltage, and current of each winding, thus determining the electromagnetic force and distribution of each winding under the load state, providing a theoretical basis for the study of transformer vibration noise [17]. Based on the electromagnetic-mechanical coupled field theory, Xuejuan Zhang et al. analyzed the electromagnetic and modal harmonic response of an 800kVA transformer winding under steady-state operation conditions and obtained the vibration characteristics of a winding undergoing loosening and deformation. They also calculated the insulation failure and verified the correctness of the simulation calculation by combining it with the winding fault vibration test [18]. Hong et al. created an assessment model that used vibrations to analyze the health of power transformers. This model analyzed the principal components of each transformer and identified the location of faults. Simulations were carried out, which proved that the model worked [19]. Zhu used ANSYS software to test the element model under normal and insulation fault conditions. They proved that the winding voltage amplitude decreased when the fault occurred and that the total harmonic distortion increased significantly [20]. Ji et al. calculated the vibration characteristics of the transformer winding under steady-state operation conditions and verified the results with test data. The results showed that with the increase in the winding preload, the fundamental frequency and higher harmonic components in the transformer vibration signal decreased [21]. To study the influence of load current on winding vibration under different mechanical conditions, Hu et al. designed transformer winding tests, analyzed the distribution of winding vibration under different winding defects, and provided a basis for detecting and analyzing the mechanical state of transformer windings [22]. Venikar et al. proposed an online method for transformer winding fault detection based on search coils, extending the application of search coils for winding displacement detection and fault identification to detect phase (U, V, and W), side (LV and HV), and the exact location of the fault. [23]. Failures in the transformer windings could also skew the signal. Different techniques for galvanically separated mains voltage waveform monitoring were assessed. The purpose of the study by Havrlik et al. was to evaluate the adequacy of each approach for estimating the active power levels as well as the degree of distortion of the output signal in relation to the input signal [24]. In order to compute the vibration displacement under various conditions, Ma Hongzhong et al. of Hohai University analyzed the vibration components and signal propagation channels on the transformer surface. They also combined the vibration mechanism and experimental analysis, built a transformer vibration model, and used the vibration signal resonance spectrum peak displacement law to pinpoint the changes in the winding preload force. In the meantime, they analyzed the fundamental frequency vibration signal's affecting the elements and suggested the fundamental frequency commutation model for winding deformation faults, which originally enabled fault type localization and classification [25].

Based on the vibration mechanism of a transformer, this paper uses COMSOL simulation software to add three modules of magnetic fields and an external circuit of solid mechanics for a three-phase three-column dry-type transformer (the transformer parameters

are shown in Table 1) through COMSOL software to calculate the vibration characteristics of transformer winding insulation, and the winding looseness, deformation, and eccentricity fault. By analyzing the fundamental frequency and higher harmonic characteristics of vibration signals under four kinds of faults, a method of a vibration increment threshold is proposed to distinguish between the typical faults of windings.

**Table 1.** Basic parameters of the transformer.

| Main Technical | Parameter | Main Technical | Parameter |
|---|---|---|---|
| Transformer model | SCB10-1000 kVA | Rated frequency | 50 Hz |
| Phase number | Three-phase | Rated capacity | 1000 kVA |
| Connection mode | Dyn11 | Rated voltage | 10/0.4 kV |
| Winding turns | 591/13 | Rated current | 57.74/1443.38 A |

## 2. Transformer Modeling

### 2.1. Transformer Model Simplification and Grid Generation

Based on the SCB10-1000/10 dry-type transformer analysis, the finite element solid model was constructed. Due to a large number of connected parts and different connection methods of transformers, it is not conducive to accurate mesh generation and finite element calculation in the later stage; the transformer needs to be simplified.

Based on the above analysis, the transformer modeling is as follows:

(1) Iron core: When building the model, the first step is to ensure that the external contour of the transformer is the same size as the real one. To be closer to reality, the core is comprised of stacked silicon steel sheets, and the silicon steel sheets are considered as a whole, so the preload of the core is not considered;

(2) Winding: The low-voltage coil is of a segmented structure, and the high-voltage is of a layered structure. The high-voltage winding is divided into four layers internally, and each layer has four layers of pads. The high-voltage winding has a support pad at the top and bottom. The low-voltage winding is divided into sections, each section is connected by four wraparound spacers, and there is also a support block above and below them;

(3) Clamping piece: the supporting fastener is relatively complex, simplifying the clamp.

According to the model of the actual dry-type transformer, the three-dimensional model of the overall transformer in Figure 1 is drawn in the geometric module of COMSOL. "A" is the high-voltage winding, and "a" is the low-voltage winding. Since the upper and lower parts are fixed and constrained symmetrically, the measuring point is selected in the upper half of the winding; for example, point 1 on the left surface of the phase A high-voltage winding. The three-dimensional structure size of the transformer body is 1575 mm × 1110 mm × 505 mm.

The overall model is divided into free tetrahedron grids. To obtain more accurate results, the winding grid was set to be conventional, the iron core was set to be coarsened, and the cushion block and brace were refined. The mesh division contains 701,423 tetrahedral elements, as shown in Figure 2.

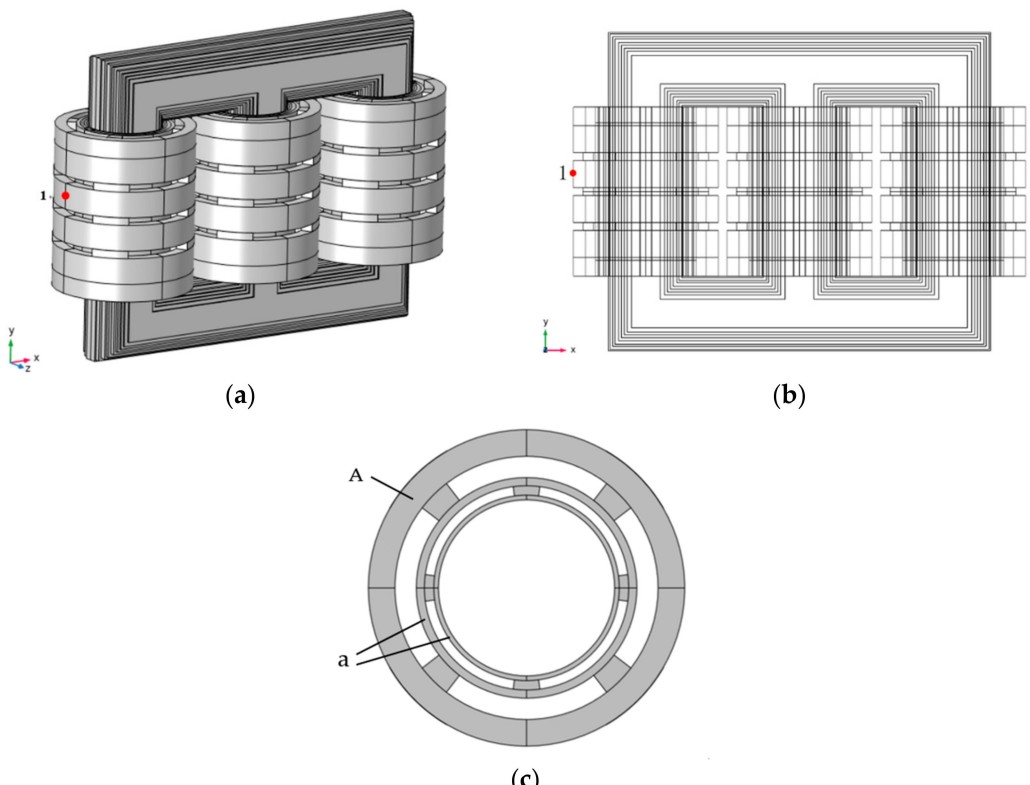

**Figure 1.** Overall model of the transformer and take point diagram. (**a**) Three-dimensional model; (**b**) a diagram for taking the points; (**c**) the top view of the winding.

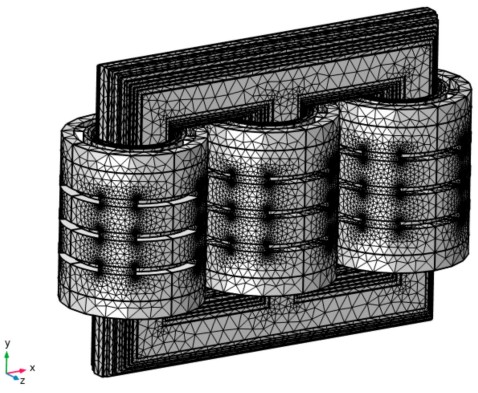

**Figure 2.** Transformer grid division.

### 2.2. Electromechanical Multiphysical Field Module

COMSOL is a finite element simulation software with powerful coupling capability. The usual process of applying COMSOL is: to select the appropriate module and partial differential equation, construct the physical model, define the material properties, impose the boundary conditions, complete the meshing, select the appropriate solver to solve, and obtain the solution results. COMSOL software was chosen because the magnetic field module, external circuit module, and solid mechanics module could be coupled more easily and efficiently. The external circuit is shown in Figure 3. Inductive loads were used in the external circuit. Based on the electric-magnetic-mechanical multi-physics coupling field theory, the vibration characteristics of the transformer were studied. The transient analysis of the transformer vibration was performed in COMSOL software by adding three modules: the magnetic field, solid mechanics, and external circuit, and the solution step of the electromagnetic field was set to 0.001 s. Additionally, the solution step of structural mechanics was set to 0.0005 s with a solution time of 0.5 s. The material is shown in Table 2.

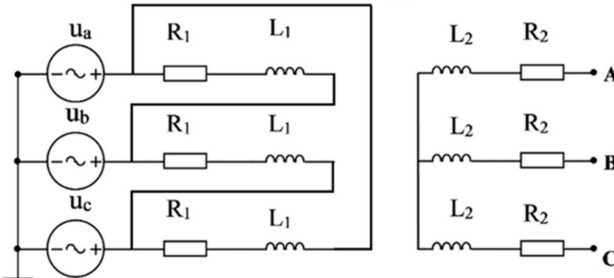

**Figure 3.** External circuit diagram.

**Table 2.** Transformer mechanical parameters.

| Parameter | Density (kg/m³) | Poisson's Coefficient | Modulus of Elasticity (Pa) |
|---|---|---|---|
| Iron core | 7870 | 0.29 | $2 \times 10^{11}$ |
| Winding | 3200 | 0.32 | $1.16 \times 10^{11}$ |
| Supporting | 1800 | 0.35 | $8 \times 10^{9}$ |
| Block | 2490 | 0.35 | $1.06 \times 10^{8}$ |

In the solid mechanics model, the winding vibration is a forced vibration under the action of the electromagnetic force, which satisfies the following dynamic equation:

$$M\ddot{s}(t) + C\dot{s}(t) + K(t) = F(t) \tag{1}$$

In Equation (1): mass; $M$ is the damping coefficient; $C$ is the stiffness coefficient; $K$ is the load vector (the electromagnetic force) applied to the winding by the model; The electromagnetic force is generated by the current in the circuit module as an excitation flowing into the magnetic field to generate electromagnetic induction. The electromagnetic field equation is:

$$\begin{cases} \nabla \times H = J \\ B = \nabla \times A \\ J = \sigma E + J_e \\ E = -\frac{\partial A}{\partial t} \end{cases} \tag{2}$$

$$\nabla \times \frac{1}{\mu} \nabla \times A + \sigma \frac{\partial A}{\partial t} = J \tag{3}$$

In Equations (2) and (3): $H$ is the magnetic field strength, $A$ is the magnetic vector potential, $J$ is the current density, $\sigma$ is the conductivity, $E$ is the electric field strength, $\mu$ is the magnetic permeability, and $J_e$ is the external current.

The multi-physical field coupling mode includes full coupling and separated coupling. The separated coupling mode has high precision and small memory occupation, so the separated coupling mode was selected as the multi-physical field coupling mode. This is shown in Figure 4.

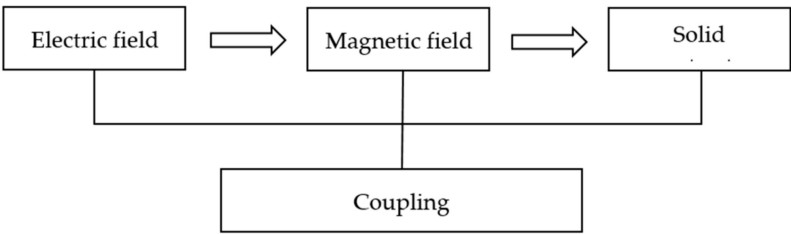

**Figure 4.** Separated coupling mode.

### 3. Modeling of the Four Faults in the Transformer Windings

*3.1. Loose Winding Fault*

For winding loosening, there is no effect on the results in the magnetic field, and the changes mainly originate in the structural force field. Therefore, consider changing the modulus of elasticity in the material parameters of the A-phase high-voltage winding to simulate the loosening of the transformer after a period of operation, after loosening, equivalent to the increase in the length of the winding in the axial direction, so slightly increase the length of each layer of the winding in the axial direction. The setting of the loosening fault is shown in Figure 5, and the changes in the mechanical parameters of the windings in the structural force field after the loosening fault are shown in Table 3.

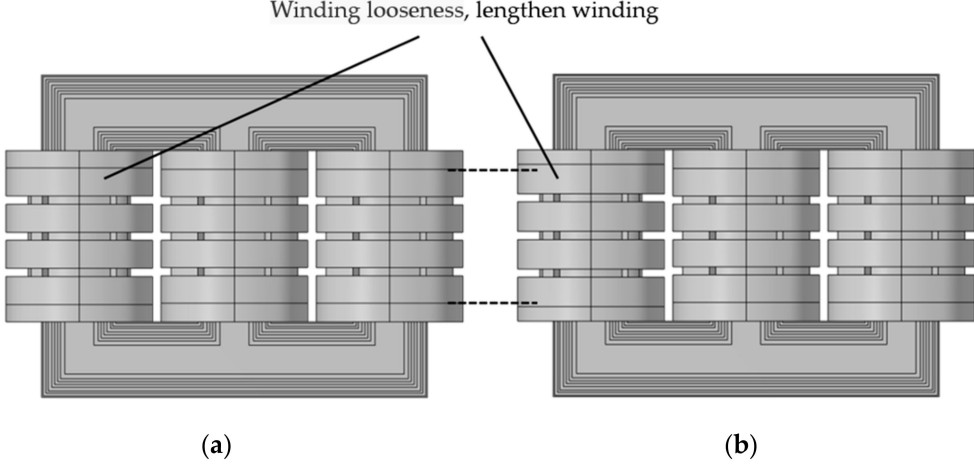

(**a**)                                                                 (**b**)

**Figure 5.** Model of the winding looseness fault. (**a**) Normal; and (**b**) loose.

**Table 3.** Parameter table of loose winding materials.

| Parameter | Normal | Looseness 20% | Looseness 40% | Looseness 60% |
|---|---|---|---|---|
| Modulus of elasticity (Pa) | $1.16 \times 10^{11}$ | $9.28 \times 10^{10}$ | $6.96 \times 10^{10}$ | $4.64 \times 10^{10}$ |

*3.2. Electromechanical Multiphysical Field Module*

For the winding insulation, the variation affects the winding vibration mainly through the magnetic field. Therefore, the effect of winding inter-turn insulation is simulated in the simulation software by reducing the number of the different turn settings of the A-phase high-voltage winding coil and the loss of the central insulation pad. The setting of the insulation fault is shown in Figure 6 and Table 4.

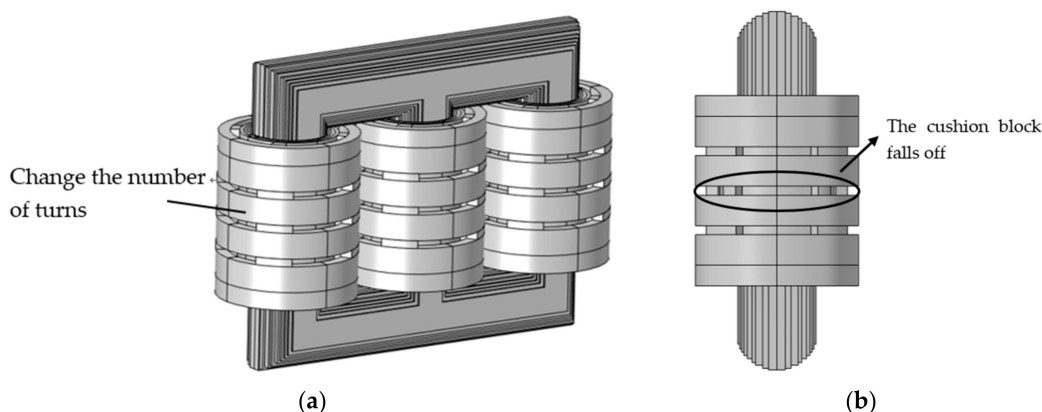

(**a**)                                                                 (**b**)

**Figure 6.** Winding insulation fault model. (**a**) Change the number of turns to simulate insulation; the (**b**) simulated insulation when the cushion block falls off.

**Table 4.** Winding insulation parameter table.

| Parameter | Normal | Insulation 1% | Insulation 3% | Insulation 5% |
|---|---|---|---|---|
| Number of turns of phase A | 591 | 585 | 573 | 561 |

### 3.3. Winding Deformation Simulation

The winding would be under the impact of a short-circuit current or bumped in transportation, which would cause the deformation of the winding. To study the displacement of the winding after deformation failure more intuitively, the focus was on the place where the deformation is large, which is the middle of the winding. A common spoke deformation is a local protrusion or the depression of the winding. Therefore, set the middle deformation of phase A high-voltage winding to 4%, 14%, and 24% of the radius to simulate the winding deformation. This is shown in Figure 7.

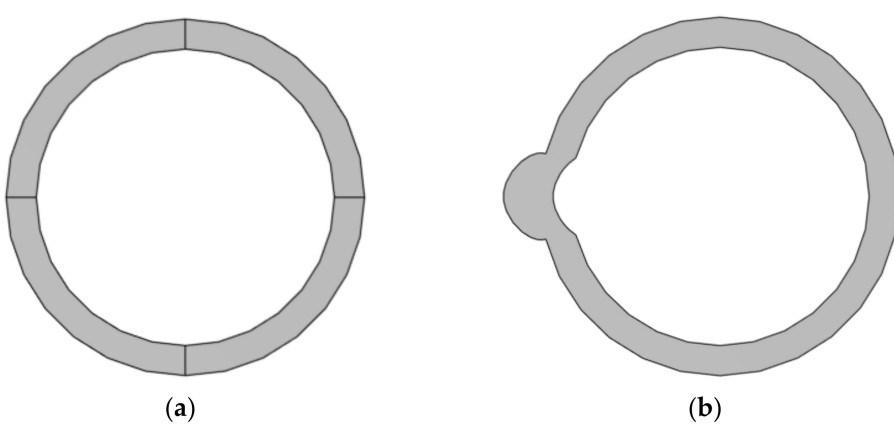

(**a**)  (**b**)

**Figure 7.** Winding deformation fault model. (**a**) Normal top view; (**b**) top view of the deformation.

### 3.4. Winding Eccentricity Fault

For A-phase and C-phase winding, under the action of the magnetic field caused by a B-phase winding current, it is easy to cause an uneven magnetic field distribution on both sides of the core column. When the electromagnetic action on the A-phase winding exceeds the mechanical deformation stress at any one place of the winding or if there is a bump, a local axis shift occurs. As there are bracing bars, which could hinder the eccentricity to a certain extent, the middle part of the high-voltage winding was set to 5%, 30%, and 60%, to simulate the eccentricity of the winding, as shown in Figure 8.

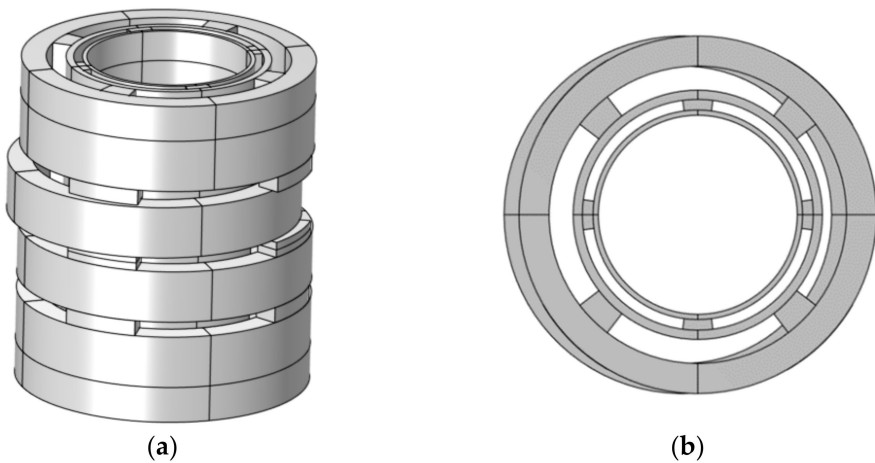

(**a**)  (**b**)

**Figure 8.** Simulation of the winding eccentricity fault. (**a**) Eccentric 3D model; (**b**) eccentric top view.

## 4. Simulation Analysis and the Identification Method of Transformer Winding Faults

### 4.1. Simulation Analysis of the Winding Looseness Fault

By analyzing the causes of winding looseness, it could be concluded that when the transformer winding looseness fault occurs, the magnetic field results obtained are basically unchanged, and the changes affecting the winding vibration mainly come from the structural force field. This is because the vibration of the transformer winding is not only affected by electromagnetic force but also closely related to the size and structure of the transformer winding geometry. Changing the shape of the transformer winding due to looseness would affect the vibration acceleration signal of the winding. To explore the influence of winding looseness on the vibration, an acceleration signal at point 1 on the winding surface was created, as shown in Figure 9.

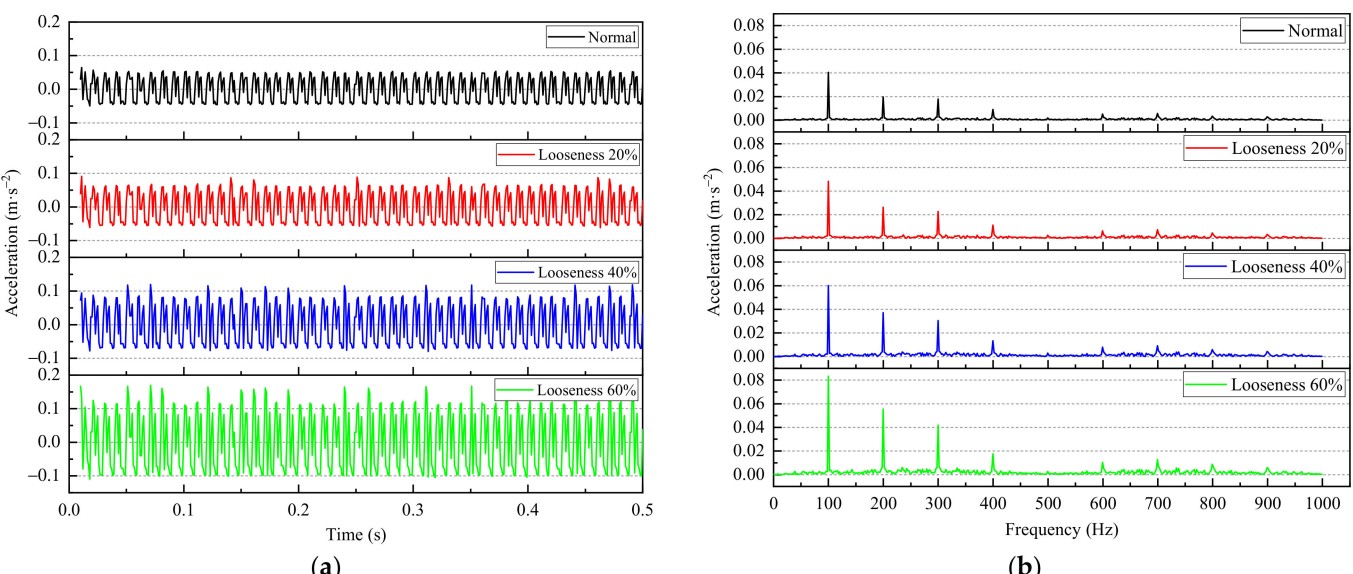

**Figure 9.** Acceleration signal at one winding surface point under normal and different looseness faults. (**a**) Time domain signal of acceleration under normal and loose working conditions; (**b**) frequency domain signal of acceleration under normal and loose working conditions.

It is seen from the above figure that when a loosening fault occurs, the acceleration increases, and the acceleration fluctuation becomes more violent with the increase in the loosening degree. After the loosening fault occurs, the frequency components in the acceleration spectrum are still dominated by 100 Hz and its integral multiple; the changes in the acceleration spectrum mainly occur at 100 Hz, 200 Hz, and 300 Hz. In order to see the changes in the acceleration spectrum more intuitively and clearly, the broken line diagrams of the acceleration spectrum amplitude changes at 100 Hz, 200 Hz, and 300 Hz were obtained, as shown in Figure 10.

It is seen from the analysis of Figure 10 that after the loosening fault occurs, the 100 Hz amplitude in the acceleration spectrum increases significantly with the aggravation of the loosening, and the acceleration amplitude increases gradually faster at 200 Hz than at 300 Hz and the 300 Hz amplitude increases more slowly. This has a similarity with the experimental results in the literature [18], which verifies the specific accuracy of the model in this paper. Among them, the amplitudes at the fundamental frequencies 100 Hz, 200 Hz, and 300 Hz are increased.

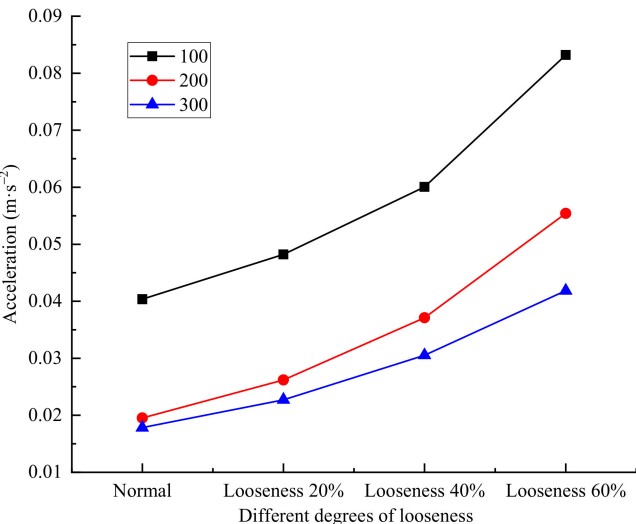

**Figure 10.** Amplitude change of loosening acceleration at 300 Hz in front of the winding.

### 4.2. Winding Insulation Fault

Therefore, for the inter-turn short circuit fault of the winding, the change mainly affects the change of the electromagnetic force through the change of the size of the electromagnetic field and then jointly affects the vibration of the winding. To explore the influence of the winding inter-turn short circuit on the vibration, the acceleration vibration signal at point 1 on the winding surface is created, as shown in Figure 11.

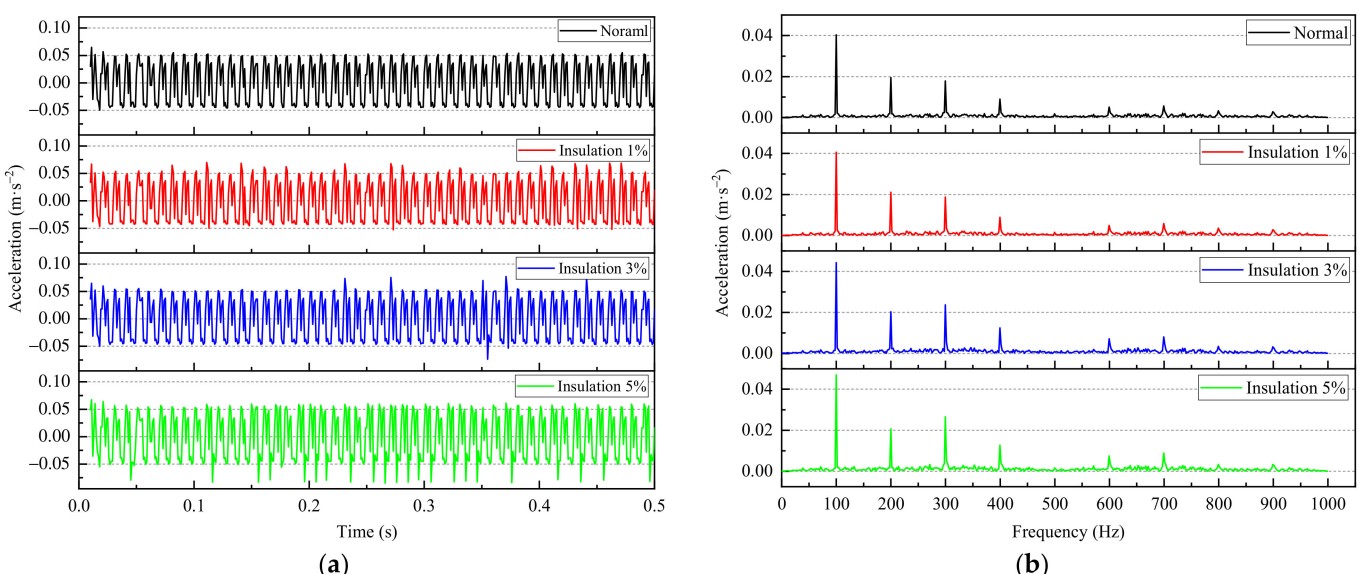

**Figure 11.** Acceleration signal under normal and different inter-turn short circuit faults at winding surface point 1. (**a**) Acceleration time domain signal under normal and insulation conditions; (**b**) frequency domain signal of acceleration under normal and insulation conditions.

It is seen from the above figures that when an inter-turn short circuit fault occurs, the acceleration increases, and with the intensification of the inter-turn short circuit, the acceleration fluctuation becomes more violent. After the inter-turn short circuit fault occurs, the frequency components in the acceleration spectrum are still dominated by 100 Hz and its integral multiple. The changes in the acceleration spectrum mainly occur at 100 Hz, 200 Hz, and 300 Hz. To see the changes in the acceleration spectrum more intuitively, the broken line diagrams of the acceleration spectrum amplitude changes at 100 Hz, 200 Hz and 300 Hz, were obtained, as shown in Figure 12.

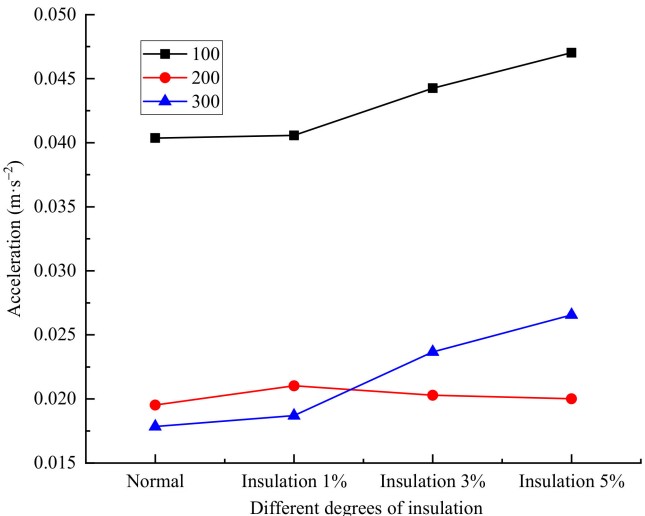

**Figure 12.** Acceleration amplitude change of different inter-turn short circuits at the first 300 Hz.

It is seen from the analysis of Figure 12 that after the turn-to-turn short circuit fault occurs, the 100 Hz amplitude in the acceleration spectrum increases slightly with the intensification of the turn-to-turn short circuit, and the acceleration amplitude increases gradually faster than 300 Hz at 300 Hz, and eventually exceeds 200 Hz. The 200 Hz amplitude changes little. This has a similarity with the experimental results in the literature [18], which verifies the particular accuracy of the model in this paper. Among them, the amplitude at the fundamental frequency of 100 Hz is not changed significantly, and 300 Hz is greater than the amplitude at 200 Hz.

### 4.3. Simulation Analysis of the Winding Deformation Fault

To explore the influence of winding deformation on the vibration, the acceleration vibration signal of point 1 on the winding surface was created, as shown in Figure 13.

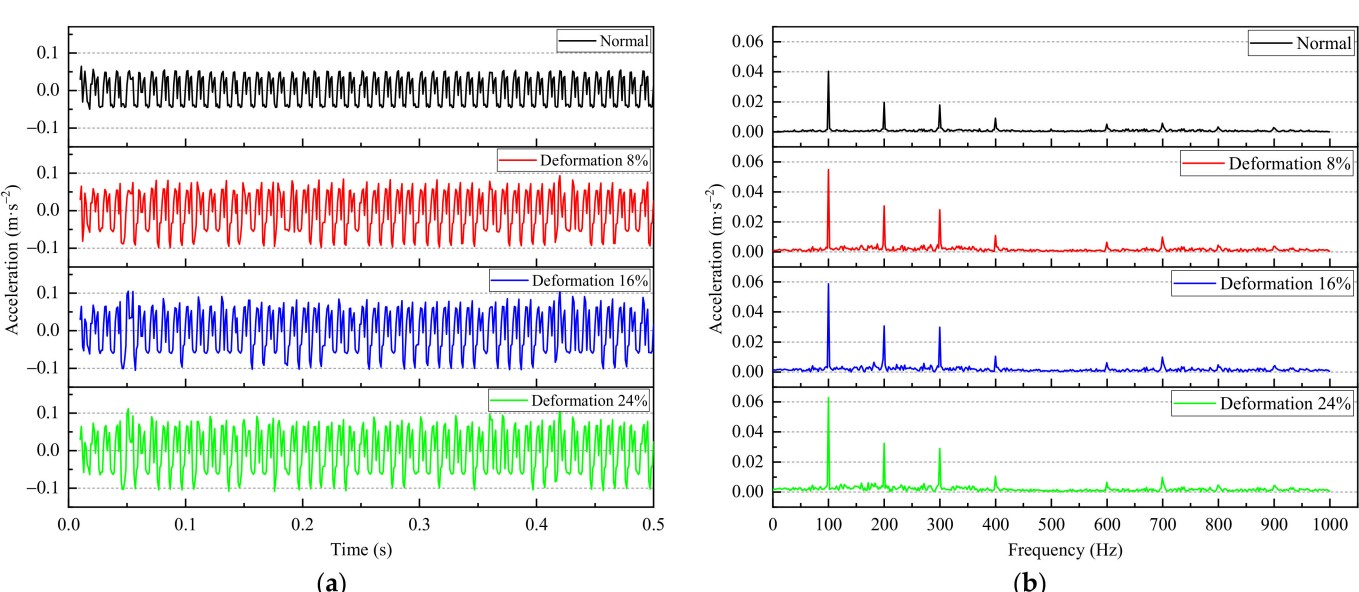

**Figure 13.** Acceleration signal at winding surface point 1 under normal and different deformation faults. (**a**) Time domain signal of acceleration under normal and deformation conditions; (**b**) frequency domain signal of acceleration under normal and deformation conditions.

It is seen from Figure 13 that when a deformation fault occurs, the acceleration increases and the acceleration fluctuation becomes more violent with the aggravation of

deformation. After the deformation fault occurs, the frequency components in the acceleration spectrum are still dominated by 100 Hz and its integral multiple. The changes in the acceleration spectrum mainly occur at 100 Hz, 200 Hz, and 300 Hz. The amplitudes at 100 Hz, 200 Hz, and 300 Hz increase with the deepening of the fault. To see the changes in the acceleration spectrum more intuitively, the broken line diagrams of the acceleration spectrum amplitude changes at 100 Hz, 200 Hz, and 300 Hz were obtained, as shown in Figure 14.

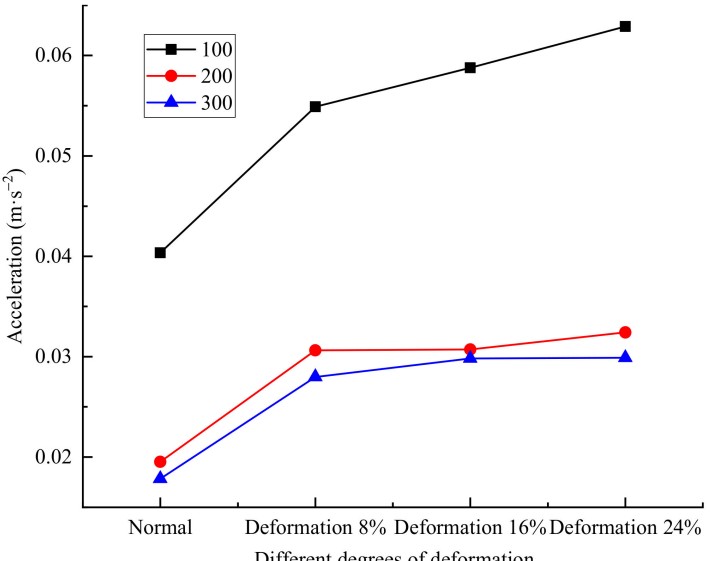

**Figure 14.** Amplitude change of the deformation acceleration at different degrees at the first 300 Hz.

It is seen from the analysis of Figure 14 that after the deformation fault occurs at the same location, the 100 Hz amplitude in the acceleration spectrum increases linearly with the deepening of the deformation, and the acceleration amplitudes at 300 Hz and 200 Hz increase slowly. The results of the different degrees of deformation, which have similarities with the experimental results in the literature [18], verify the specific accuracy of the model in this paper. Among them, the amplitude values at the fundamental frequency of 100 Hz are not changed significantly, and 100 Hz and its multiples increase significantly.

### 4.4. Simulation Analysis of the Winding Eccentricity Fault

The eccentric acceleration vibration signal of point 1 on the winding surface was obtained through calculation, as shown in Figure 15.

It is seen from the above figure that when an eccentric fault occurs, the acceleration increases and fluctuates greatly. With the increase in the eccentric distance, the acceleration amplitude and fluctuation degree are almost unchanged. When the eccentricity fault occurs, the overall eccentricity changes more violently than the local eccentricity acceleration. After a partial eccentricity fault, the change in the acceleration spectrum mainly occurs from 100 Hz to 400 Hz. After the overall eccentricity, the change in the acceleration spectrum mainly occurs from 100 Hz to 300 Hz. The amplitude of the eccentric signal in different degrees greatly increases, but the frequency components in the acceleration spectrum are still dominated by 100 Hz and its integral multiple. In order to see the change in the acceleration spectrum more intuitively, the broken line diagram of the acceleration spectrum amplitude change at the local and overall eccentricity from 100 Hz to 400 Hz was obtained, as shown in Figure 16.

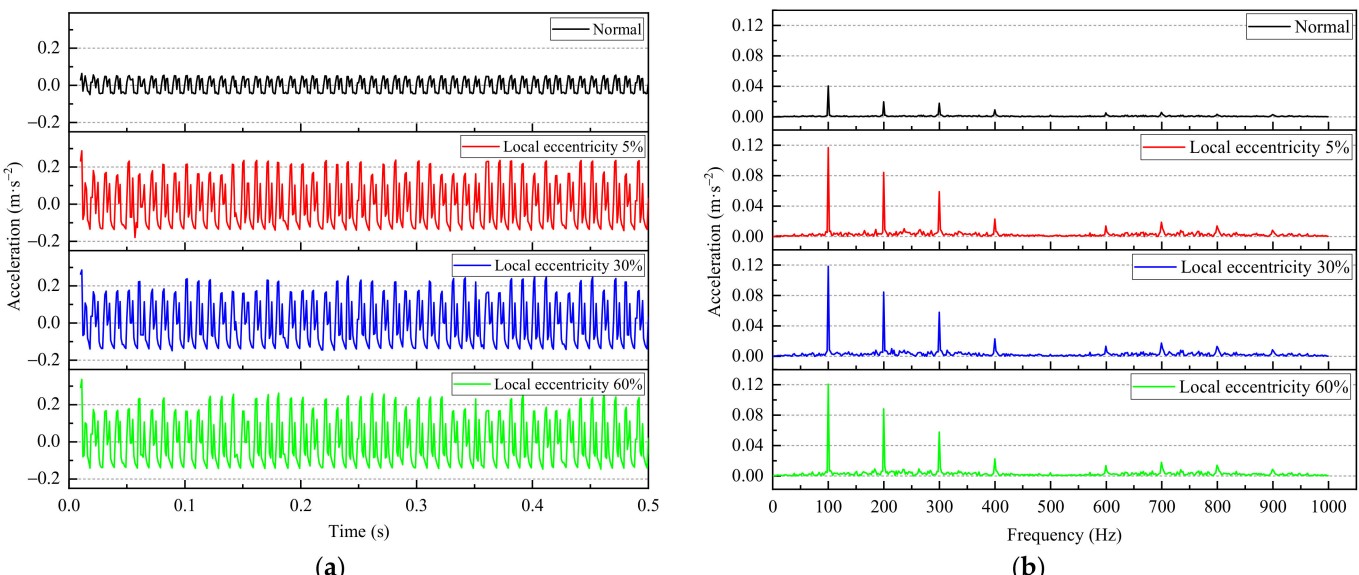

**Figure 15.** Acceleration signal at winding surface point 1 under normal and eccentric faults. (**a**) Acceleration time domain signal under normal and eccentric working conditions; (**b**) frequency domain signal of acceleration under normal and eccentric working conditions.

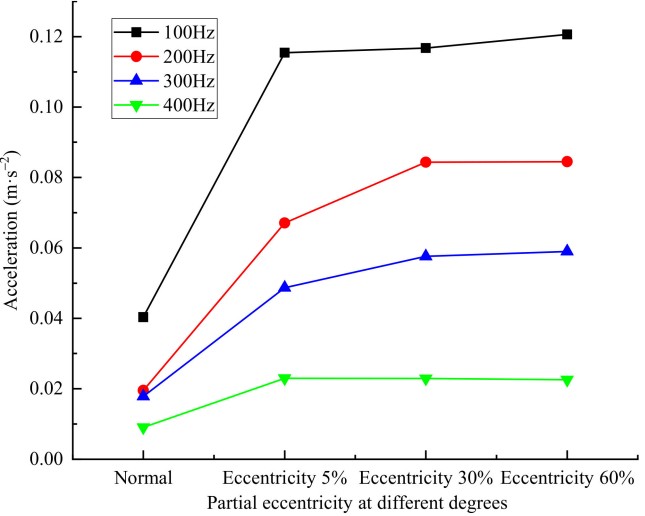

**Figure 16.** Amplitude change of eccentric acceleration in different degrees at the first 400 Hz.

It is seen from the analysis of Figure 16 that the amplitude from 100 Hz to 400 Hz increases sharply after the eccentric fault occurs, but the acceleration caused by local eccentricity increases slightly with the deepening of the fault. This is because the eccentricity is far away from the B-phase winding and the C-phase winding to a certain extent, and the magnetic field of the local eccentricity decreases one layer at a time. The reduced amplitude is smaller than the interference from the magnetic field of the whole A-phase winding, so it is slightly increased.

### 4.5. Winding Fault Identification Method

How to distinguish these four kinds of faults is a key problem in transformer vibration fault diagnosis. After the coupling calculation of multiple physical fields, the vibration characteristics of transformers under different faults were obtained. The characteristics of the x-direction vibration signals of the same measuring point 1 on the transformer under normal conditions were compared, as shown in Figure 17.

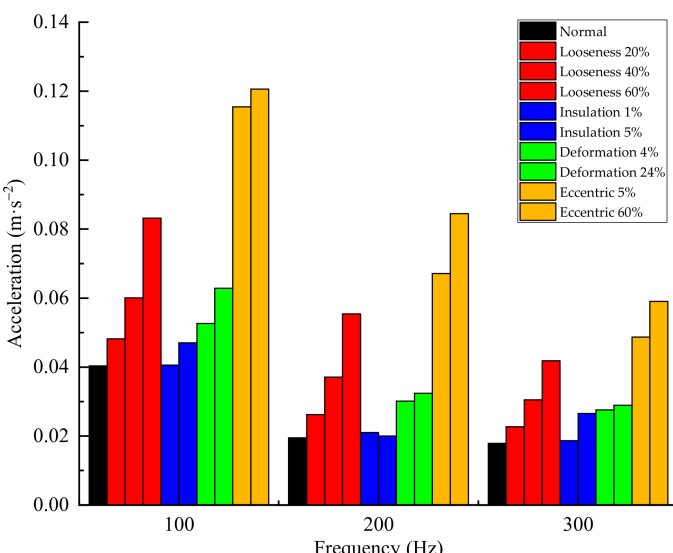

**Figure 17.** Acceleration frequency domain signal under different faults at the first 300 Hz winding surface point 1.

It is clearly seen from Figure 17 that when the winding is loose, eccentric, and deformed, the acceleration signal increases to a certain extent as the degree increases. When an insulation fault occurs, it is clearly seen that the amplitude of 300 Hz is higher than 200 Hz; each fault changes within its own range at the first 300 Hz.

Figure 17 only depicts the approximate acceleration amplitude change of each fault and does not accurately distinguish winding faults. In order to distinguish the faults of different windings with vibration signals more easily, a method of a vibration increment threshold is proposed. Since the acceleration growth gradient of each fault is different, the vibration increment threshold (the acceleration growth rate) is introduced to determine the winding fault:

$$K_i = \frac{A_i - A_{Ni}}{A_{Ni}} \tag{4}$$

wherein, $A_i$ represents the amplitude of winding vibration acceleration under fault conditions and $A_{Ni}$ represents the amplitude of winding vibration acceleration under normal conditions (*i* represents 100 Hz, 200 Hz, and 300 Hz of windings). Through this calculation, the fundamental frequency, double frequency, and triple frequency values of a transformer winding are obtained, as shown in Table 5.

**Table 5.** Fault characteristic value $K_i$ at winding surface point 1.

| Working Condition | Looseness | Insulation | Deformation | Eccentric |
|---|---|---|---|---|
| $K_1$ (100 Hz) | 19~106 | 1~17 | 30~45 | 189~199 |
| $K_2$ (200 Hz) | 34~183 | 2~8 ↓ | 51~54 | 332~333 |
| $K_3$ (300 Hz) | 27~135 | 5~49 | 45~57 | 223~231 ↓ |

By analyzing Table 5 (the arrow represents that the fault decreases with the deepening of the fault) and Figure 18, we could conclude the judgment rule of the transformer winding: when the fault is initially formed because the fault characteristics are not obvious, it is impossible to distinguish which fault has occurred. When the fault starts to increase, each fault has certain characteristics, which could be distinguished by the $K_1$ value first and combined with the $K_2$ sum $K_3$ value-assisted method. When $K_1$ is in the range of 1 to 18, the insulation fault of the winding could be determined by combining the signal characteristics of the insulation fault at 300 Hz greater than at 200 Hz. When the $K_1$ value is greater than 180, it is determined that the winding has an eccentric fault. When the $K_1$ value is

within the range of 18 to 180, it is impossible to determine whether there is looseness or deformation. Because $K_1 K_2 K_3$ corresponds to the fault, the corresponding loose $K_2$ value could be calculated according to the $K_1$ value falling within the deformation $K_1$ value range when loosening to determine whether it falls within the deformation $K_2$ value range and then determine which fault occurs. In this paper, when a 30% looseness fault occurs, the $K_1$ value is 32, which falls between the deformation, but the $K_2$-calculated looseness value of 67 is far greater than the range of deformation; therefore, the $K_2$ value ranges from 51 to 54. Therefore, when the $K_1$ value is 32, a loosening fault occurs.

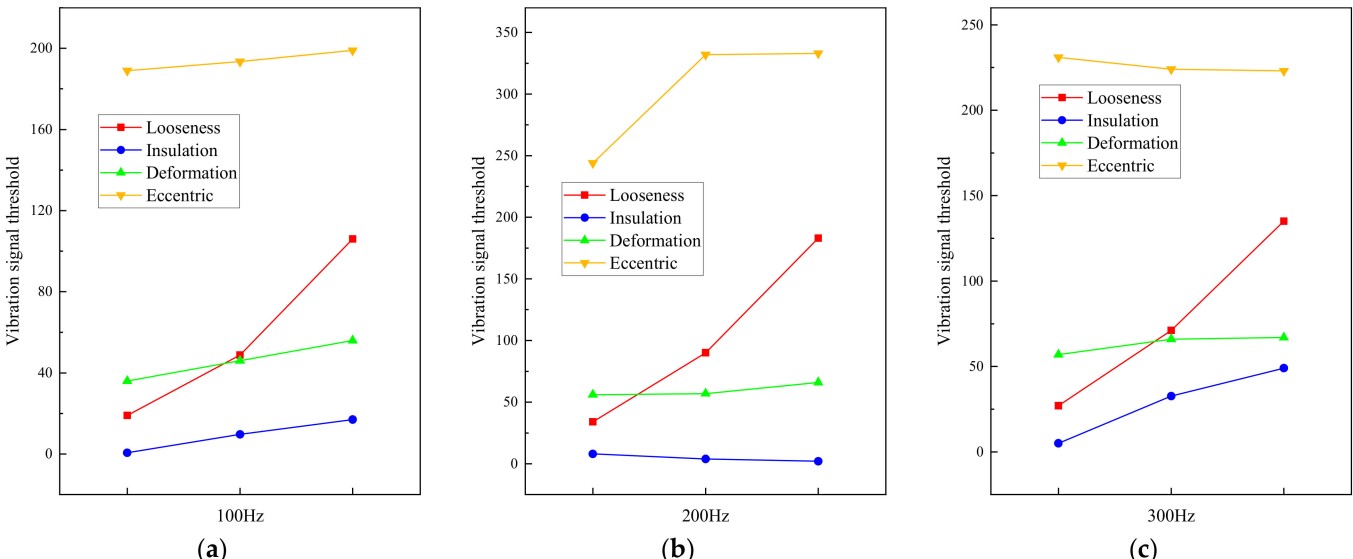

**Figure 18.** Broken line diagram of the vibration increment threshold $K_i$ at point 1 on the winding surface. (**a**) Line chart of the vibration increment threshold at 100 Hz; (**b**) line chart of the vibration increment threshold at 200 Hz; (**c**) line chart of the vibration increment threshold at 300 Hz.

## 5. Conclusions

In this paper, COMSOL was used to simulate four typical winding faults. By analyzing the fundamental frequency and harmonic frequency characteristics of vibration fault signals, a vibration increment threshold method for identifying four winding faults is proposed. The following conclusions are drawn:

(1) When the threshold $K_1$ value of the fundamental frequency is within the range of 1 to 18, the winding likely has an insulation fault, which could be judged without using $K_2$ and $K_3$. When the threshold $K_1$ value of the fundamental frequency exceeds 180, the winding may have an eccentric fault.

(2) When the threshold $K_1$ value of the fundamental frequency is within the range of 18 to 180, looseness or deformation faults may occur. At this time, the double frequency $K_2$ and triple frequency $K_3$ could be combined to calculate $K_2 K_3$ whether it falls within the corresponding fault range to determine which fault may occur.

This study provides a reference for transformer winding fault diagnosis.

**Author Contributions:** Conceptualization, Writing—original draft, Writing—review & editing, Supervision, Methodology, Validation, S.L.; Conceptualization, Writing—original draft, Methodology, Software, L.Z.; Funding acquisition, Writing—review & editing, Project administration, L.Y.; Resources, Formal analysis, Investigation, Visualization, C.G. and Z.W. All authors have read and agreed to the published version of the manuscript.

**Funding:** This research was funded by the National Natural Science Foundation of China (51175474).

**Conflicts of Interest:** The authors declare no conflict of interest.

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
