# Peer review of "The Identification Method of the Winding Vibration Faults of Dry-Type Transformers"

_electronics, doi:10.3390/electronics12010003_

Round 1

Reviewer 1 Report

The article deals with current issues. Diagnostika poruchy vinutí transformátoru je důležitá pro pÅ™enos elektrické energie. Reading the article was enlightening for me as well. I propose to accept the article after minor revision, see comments.

Comments:

- Table 1 – Probably not „Main Techinical“, but „Main Technical“ (two times).

- Line 110 – „1575 × 1110 × 505 mm“ the unit have no physical sense. There must be  „1575 × 1110 × 505 mm3“, or „1575 mm × 1110 mm × 505 mm“.

- Lines 106, 116, 144, 165, 221, 247, … - Why are the image numbers in blue font?

- Table 2 - There should be a space in front of 20%, 40%, 60%. The values should be better 1.16 × 1011  than 1.16E+11, …

- Table 3 - There should be a space in front of 1%, 3%, 5%.

- Figures 8, 9, 11, … – There should be better „Acceleration (m.s-2)“ than „Acceleration (m/s^2)“.

- For example, the article doi:10.3390/s22207769 deals with signal distortion in electronic inverters. Perhaps it could be cited in introduction, because probably transformer winding faults will also cause signal distortion.

Reviewer 2 Report

General information

The article presents a model for the simulation of power transformer defects that can be diagnosed using the vibroacoustic method. On the basis of a real transformer, a simulation model was prepared, in which vibroacoustic signals from 4 different system defects were modeled.

In general, the subject matter certainly coincides with the subject of the journal and has an appropriate scientific level. The authors chose the right analysis tools and performed the research correctly. However, a few important issues were missing, which are discussed below.

In terms of editing and language the article is of a sufficiently standard. The article also uses graphics of appropriate quality.

Remarks and question for discussion

11. In 107 line is a sentence” A is the high-voltage winding and a is the…”. It is not  clear what is the “A” and “a”, because there are not showed in the Figure 1. Besides it would be better if signs A and a will be written in quotation marks.

22. The caption under figure 2 has moved to page 4.

33. In Figure 4, the images for the winding looseness defect are marked. In the graphic, it is impossible to see any difference in pictures a and b (normal and looseness), so I don't know what is the point of showing such a graphic?

44. The model was based on a real object, but it was not verified on it in any way. Why? This would be the starting point for later modifications of the simulation model.

55. In the work, some characteristic waveforms and system responses to modeled defects were selected. I understand that it was not possible to compare the results with similar defects in the real system. But why did the authors not compare the results with the literature data? I think you can find frequency ranges of vibroacoustic signals recorded in typical transformer defects. Otherwise, it is difficult to assess to what extent the created model corresponds to reality.

56. The work analyzed defects acting separately. In fact, these defects may overlap, giving a completely different picture (perhaps not a superposition). Have the authors done this type of analysis?

77. Among the modeled defects, I missed a rather characteristic defect, which is the buckling of the winding in the Y axis. Do the authors consider modeling such a defect as well?

Reviewer 3 Report

Line 44, Line 47, Line 53, Line 57, Line 62, Line 66, Line 71, Line 73; the authors use different types of citations. They use IEEE citation: [1] and non-standardized citation: Zhang et al. It is convenient to use only one type of citation.

//****//

In line 56-57, the sentence: " ..., providing a basis for the study of transformer vibration noise Theoretical basis[14]", is a bit difficult to understand.

//****//

Figure 1. It is very difficult to see the POINT 1 in the figure.

//****//

What type of load is used in the external circuit? It is very important because a capacitive load has a different behavior than a resistive-inductive load. The authors should indicate this.

//****//

A remark: J is not current. J is current density. Current I[A]. Current density J[A/m2].

Current = integral of[ J·dS]. Current is a scalar. J is a vector. dS is differential surface vector.

//****//

Line 130, Equation (1), what are the values of M and C?

//****//

Figure 4. It is difficult to distinguish winding looseness in the figure. Dotted line indicate that, but, at a glance, it is not seen. The authors should explain that into the text.

//****//

There are winding vibrations, iron core vibrations (magnetostriction) and vibrations produced by cooling equipment. Have the authors considered the overlaps of these types of vibrations?

//****//

In Table 4, for Looseness, the range of K1, K2 and K3 is within the range of K1, K2 and K3 for Deformation. How is it possible to distinguish a looseness fault from a deformation fault?

For example: K1 Looseness equals 25, and K1 Deformation equals 25; K2 Looseness equals 52, and K2 Deformation equals 52; K3 Looseness equals 46, and K3 Deformation equals 46;

//****//

Authors should make a brief reference to COMSOL (web or similar). Not everyone knows COMSOL software and how it works.

//****//

Reviewer 4 Report

English writing should be highly improved. There are several errors.

More details of your simulation model  in COMSOL should be presented.

Details of materials properties which you implemented in the COMSOL model should be presented.

You said in introduction section, dry type transformers have small size. Is it true? In my opinion, oil immersed transformers are smaller than dry type transformers. What do you think about this?

Your literature review should be improved highly.

In my opinion a paragraph should be added in the introduction section. In this paragraph mechanical concerns of the transformer windings should be reviewed. For example short circuit forces and inrush current forces should be reviewed. For example three references in this topic which are published in the last three years are:

“HTS Transformers Leakage Flux and Short Circuit Force Mitigation through Optimal Design of Auxiliary Windings,” Cryogenics, vol. 110, September 2020, 103148.

“Optimal Design of Flux Diverter Using Genetic Algorithm for Axial Short Circuit Force Reduction in HTS Transformers,” IEEE Transactions on Applied Superconductivity, vol. 30, no. 1, pp. 1-8, January 2020.

“Multi-Segment Winding Application for Axial Short Circuit Force Reduction under Tap Changer Operation in HTS Transformers,” Journal of Superconductivity and Novel Magnetism, vol. 32, no. 10, pp. 3171-3182, October 2019.

Round 2

Reviewer 2 Report

Thank you for the discussion and for answering my questions. After taking into account the corrections made, the article can be published.

Reviewer 4 Report

in my opinion, now this manuscript may be published.